# Measuring Reactive Oxygen Species in Semen for Male Preconception Care: A Scientist Perspective

**DOI:** 10.3390/antiox11020264

**Published:** 2022-01-28

**Authors:** Patience E. Castleton, Joshua C. Deluao, David J. Sharkey, Nicole O. McPherson

**Affiliations:** 1Freemasons Centre for Male Health and Wellbeing, The University of Adelaide, Adelaide 5005, Australia; patience.castleton@adelaide.edu.au (P.E.C.); joshua.deluao@student.adelaide.edu.au (J.C.D.); 2Robinson Research Institute, The University of Adelaide, Adelaide 5005, Australia; david.sharkey@adelaide.edu.au; 3Adelaide Health and Medical School, School of Biomedicine, Discipline of Reproduction and Development, The University of Adelaide, Adelaide 5005, Australia; 4Repromed, 180 Fullarton Rd., Dulwich 5065, Australia

**Keywords:** fertility, sperm, seminal plasma, ART, pregnancy

## Abstract

Oxidative stress and elevated levels of seminal and sperm reactive oxygen species (ROS) may contribute to up to 80% of male infertility diagnosis, with sperm ROS concentrations at fertilization important in the development of a healthy fetus and child. The evaluation of ROS in semen seems promising as a potential diagnostic tool for male infertility and male preconception care with a number of clinically available tests on the market (MiOXSYS, luminol chemiluminescence and OxiSperm). While some of these tests show promise for clinical use, discrepancies in documented decision limits and lack of cohort studies/clinical trials assessing their benefits on fertilization rates, embryo development, pregnancy and live birth rates limit their current clinical utility. In this review, we provide an update on the current techniques used for analyzing semen ROS concentrations clinically, the potential to use of ROS research tools for improving clinical ROS detection in sperm and describe why we believe we are likely still a long way away before semen ROS concentrations might become a mainstream preconception diagnostic test in men.

## 1. Introduction

Cases of infertility are increasing worldwide, affecting an estimated 48.5 million couples every year [1]. Of these couples, approximately 30% of infertility diagnoses are attributable solely to a male factor, with male infertility contributing to up to 50% of all cases [2]. There is a growing call for men to be actively involved in preconception care [3], however despite their equal role in conception there is currently a disparity in preconception care and testing available to men. One of the biggest limitations in male preconception testing is that while the current standard clinical diagnostics for male factor infertility, being sperm count, motility and morphology, reveal useful information for the initial evaluation of male infertility, it is not a direct test of fertility [4]. Importantly, in patients with normal sperm parameters (normospermic), these sperm parameters alone are not predictive of pregnancy following assisted reproductive treatment (ART) [5]. In addition, recent studies have argued that the lack of standardization in the way routine semen analyses are performed across clinics, often absent of appropriate quality control measures, may limit the accuracy of test interpretations [6]. This can lead to lengthy, stressful, and costly processes for diagnosing male infertility/subfertility, which in turn reduces the likelihood of men to undertake further preconception fertility testing [7].

Oxidative stress and elevated levels of seminal and sperm reactive oxygen species (ROS) have been reported as a contributing factor in up to 80% of all infertility diagnoses [8], with important consequences for the generation of a healthy conceptus and offspring observed in animal models [9]. This highlights its potential usefulness not only for diagnosing male subfertility/infertility, but also the possibility of holding predictive value on the likelihood of a couple having a healthy pregnancy and child. While evaluating ROS concentrations in semen shows promise as a potential diagnostic tool, no current mainstream guidelines have been established informing which individuals to test, nor which of the various tests to perform [10]. Although, we do note proposed guidelines have been published for the management of male oxidative stress in idiopathic male infertility [11]. 

Further, normal reference ranges (or newly introduced ‘decision limits’) for ROS concentrations in semen are conflicting (see Table 1). As a result, it is suggested in the most recently updated (2021) 6th edition of the WHO laboratory manual for the examination and processing of human semen that the diagnostic predicative values of ROS still need to be interpreted with caution [12]. These limitations have largely contributed to the reduced clinical usage of ROS measurement within the diagnostic laboratory. In this review, we provide an update on the current techniques used for analyzing semen ROS concentrations clinically, the potential to use of ROS research tools to improve clinical ROS detection in sperm and describe why we are likely still a long way away before semen ROS concentrations become a mainstream preconception health test in men. 

## 2. The Biology of ROS in Sperm

Reactive oxygen species are an oxygen containing molecule that has a free electron or unstable bond and it is these characteristics that allow the molecule to react with nucleic acids, lipids, proteins, and carbohydrates within cells. An over production of ROS has been implicated in the development of aging and many chronic and degenerative diseases including cancer, chronic obstructive pulmonary disease, cardiovascular disease, neurodegenerative disorders (Alzheimer’s, Huntington’s Parkinson’s, amyotrophic lateral sclerosis, spinocerebellar ataxia, ischemic stroke) and inflammatory bowel disease [13,14,15]. Many forms of ROS are found in sperm including superoxide (O_2_^−^), hydrogen peroxide (H_2_O_2_), hydroxyl radical (OH), peroxyl (ROO^−^), singlet oxygen (O_2_), nitric oxygen (NO), peroxide ion (O_2_^2−^) and hydroxyl ion (OH^−^) all which have different biological targets with their own spectrum of reactivity [16,17]. ROS generation in sperm and seminal plasma can occur (i) endogenously as a by-product of aerobic metabolism and ATP production in sperm mitochondria and the nicotinamide adenine dinucleotide phosphate oxidase pathway (NOX) in sperm plasma membranes or (ii) exogenously from sources such as but not limited to elevated leukocytes in seminal plasma in response to genital tract infection or inflammation, smoking, excessive alcohol consumption, exposure to radiation, genital heat stress and obesity [18,19,20] (Figure 1). A biological concentration of ROS must be present in sperm for normal physiological process such as capacitation, hyperactivation and the acrosome reaction to occur allowing sperm to be primed for successful fertilization. ROS facilitates capacitation through its activation of adenylyl cyclase which then acts to convert ATP into cyclic AMP (cAMP). cAMP’s downstream effects result in the phosphorylating of proteins required for hyperactivation which is assisted by ROS inhibition of tyrosine phosphatases [17,21,22]. ROS is also required for the acrosome reaction and fusion of the gametes, as it oxidizes and extrudes cholesterols within the membrane, thereby increasing its fluidity [23]. However, at either end of the spectrum (either too low or too high) ROS concentrations in sperm can be detrimental.

A low levels, ROS inhibits sperm capacitation via reduced activation of adenylyl cyclase, while at high levels, ROS induces sperm lipid peroxidation and DNA damage [23]. Sperm are highly susceptible to ROS damage due to their lack of cytoplasmic scavenging enzymes and high levels of polyunsaturated fatty acids found in their plasma membranes which reduces their capacity to repair ROS related damage [24]. Sperm fatty acid membranes contain relatively unstable bonds that can be readily oxidized by ROS to create lipid radicals that go on to react with nearby fatty acids in a self-perpetuating cycle [25,26] to form lipid peroxidation. The degradation of sperm membranes especially in the midpiece leads to a decrease in sperm motility which is a hallmark of male infertility [27]. ROS related sperm DNA damage occurs through oxidization of guanine into 8-hydroxy-2′-deoxyguanosine (8OHdG). Sperm can excise this base using base excision repair mechanisms, but they only possess half of the necessary machinery, with the other half of this machinery is present in the oocyte [28]. Therefore, this abasic site will remain unrepaired until fertilization or, detrimentally, it can destabilize the DNA molecule leading to single strand breaks, double strand breaks and ultimately DNA fragmentation [17], making any ROS induced DNA damage in sperm able to be transmitted at conception [24]. This can result in mutagenesis, peroxidation of unsaturated lipids, disruption of mitochondrial metabolism and alterations to methylation marks in the early embryo leading to heritable mutations and pediatric phenotypes [29]. 

Elevated ROS concentrations in sperm has been observed in male pathologies including obesity, smoking, excess alcohol intake, aging, environmental and occupational exposure, and subfertility, all which have been associated with subfertility, increased offspring susceptibility to disease and reduced lifespan in men [30]. ROS concentrations have been proposed as being as a useful biomarker for fertility testing in male preconception care. However, the optimal concentration of seminal/sperm ROS is not yet fully understood, despite several clinical products being commercially available. 

## 3. Current Clinical Measures of ROS in Semen

Two of the three current, clinically available measurements of ROS in semen, being Luminol and MiOXYSIS, have been added to the advanced examinations section in the most recent 6th addition WHO manual for assessment of human semen [12]. However, it is noted that from ‘a clinical diagnostic perspective, this group of assays should only be used and interpreted with caution until more conclusive proof of their diagnostic relevance exists’. Currently, ROS measures tend only to be performed in men if abnormalities are observed during standard semen analysis, or when couples undergoing ART have experienced reduced fertilization and embryo development rates, or repeated implantation failure [8]. The three current commercially available tests are discussed below, with their strengths and limitations presented in Table 1 and all relevant cohort studies presented in Table 2. 

### 3.1. Luminol Chemiluminescence

The luminol-dependent chemiluminescence method involves a sensitive luminescent probe which reacts with various free radicals in whole semen, including H_2_O_2_, O_2_^−^ and OH^−^, that allows both intra- and extracellular ROS to be measured on a luminometer [61]. There are many luminometers on the marker that can be used to measure the intensity of the light produced during the reaction one of two ways (i) photon counting expressed as relative light units (RLU) or (ii) electric current expressed as counted photons per minute (cpm) or mV/s [62]. Single and double tube luminometers are relatively inexpensive (~$10,000 AUD), however, can only measure one or two samples a once, while multiple tube luminometers are more expensive (~$30,000 AUD) but can measure multiple samples at once [62]. Further, there are also plate luminometers for large sample turn over, however, a limitation of this type of set up is that plates need to be disposed of after opening even if measuring only one sample [62]. There have been a number of cohort studies trying to determine clinical reference ranges of luminol for the diagnosis of male infertility (Table 1). One of the first cohort studies performed in 2007, assessing 144 fertile men seeking vasectomy and 47 subfertile men found that a cut off of 5 × 10^3^ cpm/20 × 10^6^ sperm was able to determine difference between fertile and infertile men with a specificity of 76% and sensitivity of 73.3% [33]. A study by Desai et al. in 2009 using a similar device in 54 infertile men and 51 fertile donors established a cut off of 1.85 × 10^4^ cpm/2 × 10^7^ sperm with 77.8% sensitivity and 82.4% specificity [35]. In machines assessing RLU, similar differences in reported reference ranges have been found. Agarwal et al. in three separate studies comparing fertile donors vs infertile men established three different cut offs (1) 91.9 RLU/sec/10^6^ sperm (sensitivity 93.8%, specificity 68.8%) [40], (2) 102.2 RLU/sec/10^6^ sperm (sensitivity 76.4% and specificity 53.3%) [39] and (3) 93 RLU/sec/10^6^ sperm (specificity of 70.4% and sensitivity of 61.4%) [37]. Differences in established clinical cut offs between studies could be due to one of the major limitations of luminol, which is its inability to distinguish ROS produced from sperm and other cell types within the ejaculate (including immature sperm and leucocytes). Athayde et al. found that clinical cut offs were much higher (0.5 vs. 1.25 × 10^4^ cpm/2 × 10^7^ sperm) in men with leucospermia [33], and therefore, reference ranges need to account for the presence of these cell types. Another limitation of luminol which may have hindered its clinical roll out is that measurements in semen are not stable over time and start declining immediately after ejaculation [41]. This may be due to the instability of the reagents, with luminol easily oxidized in the presence of light and the assay sensitive to temperatures above 25 °C [41]. Although, the same study reported working reagents were stable for 3 months when stored at correct temperatures and protected from light [41]. Further, luminol requires quite a large sample volume (400 µL semen) for analysis, which is likely fine for men undergoing standard semen analysis, however, may be problematic for measurements occurring in samples prior to use in ART. Currently, only one study has assessed semen ROS as measured by chemiluminescence on pregnancy outcomes in 147 infertile men undergoing in vitro fertilization (IVF) or intracytoplasmic sperm injection (ICSI) [42]. 43% of men were found to have high ROS levels (above 10 mV/sec/10^9^), with high levels of ROS found in men whom partner did not get pregnant. Interestingly, this relationship between increased semen ROS and negative pregnancy was only present in couples undergoing IVF and not ICSI. This outcome was influenced by a high level of basis as only 11 out of the 41 couples undergoing IVF had a positive pregnancy, data was analyzed by simple statistics (an independent-samples *t*-test) and results were not adjusted for other clinical or biological factors that could influence outcomes [42]. Therefore, studies evaluating the effectiveness of luminol for determining ROS concentrations in semen and its potential relationship with pregnancy and live birth outcomes in ART are warranted. 

### 3.2. MiOXSYS^TM^

The MiOXSYS System™ (Englewood, CO, USA) is a highly specific in vitro diagnostic tool used to measure static Oxidation-Reduction Potential (sORP) in human semen [45]. Of all the currently available clinical measures of ROS, the MiOXSYS System™ has been one of the most extensively researched recently (Table 1). The system works by measuring the transfer of electrons between oxidants and reductants within fresh semen samples to ultimately calculate total oxidant and antioxidant activity present within the ejaculate. The assay requires only 30 µL of fresh or frozen semen sample and produces results in less than 5 min, making it a popular choice in both clinical and research settings. While the MiOXSYS™ reader is relatively low cost (~$3000 AUD), the sensors required for testing are quite expensive in comparison (~$65–80 AUD). In the largest cohort analysed to date (2092 men attending semen analysis from 9 countries), Agarwal et al., found that the assay yields high sensitivity and positive predictive values (98.1% and 94.7% respectively) for male infertility when assessing based on abnormal semen parameters, but much lower specificity and negative predictive values (40.6% and 66.6% respectively) of fertile men [49]. They found that sORP levels were negatively correlated with sperm concentration, motility and morphology [49], with normed sORP values needing to be adjusted for sperm count and a value of 1.36 mV/10^6^ able to differentiate fertile and infertile men [49]. In contrast, a previous large cohort study assessing 1168 infertile men and 100 fertile men reported higher cut-off values, with 1.73 mV/10^6^ required for determining oxidative stress when analysing semen samples with multiple semen defects [43]. Further, a recent study assessing the normal physiological sORP range by incubating semen samples with cumene hydroperoxide and ascorbic acid found normal sperm function was present at sORP ranging between −9.76 and 1.48 mV/10^6^ [51] and therefore cut offs of >1.48 mV/10^6^ should be applied. The MiOXSYS System™ also has many protocol advantages. It is not heavily influence by assay times, with similar sORP measurements present even after 2 h post ejaculation [45], alleviating time constraints, that normally exist within other ROS measurement methods (i.e., Luminol) or performing of standard semen analysis [12]. MiOXYS^TM^ can be used for assessments of static oxidation-reduction potential in seminal plasma [45] and sperm preparation media, freezing media and embryo culture media [50]. It exhibits good reproducibility across operators, analysers and days and results are not affected by mechanical agitation or snap freezing/thawing [47]. One potential limitation is that measurements are temperature sensitive (2–37 °C) with levels significantly increasing outside of this range [47]. While the MiOXSYS System™ has been evaluated in a number of large cohorts of infertile men, we could find only one published study to date (50 idiopathic infertile couples) that extended their analysis to evaluate fertilization and pregnancy outcomes, finding a cut off sORPmV < 1.57 mV/10^6^/mL predictive of fertilization and < 0.75 mV/10^6^/mL for clinical pregnancy [52]. Further large-scale cohort studies extending their analysis into pregnancy and live birth outcomes are still needed to determine the clinical utility of using MiOXSYS™ for ART.

### 3.3. OxiSperm^®^

OxiSperm^®^, produced by Halotech^®^, measures the presence of excess O_2_^−^ in sperm, seminal plasma and whole semen. The assay is largely based on nitro blue tetrazolium (NBT), in which the yellow NBT molecule is reduced into the insoluble blue crystals, called formazan, in the presence of O_2_^−^ [63]. This can be visualized either in the form of a reactive gel, in the instance of OxiSperm^®^II (Madrid, Spain), or through their presence in sperm/leucocytes under bright-field microscopy (OxiSperm^®^), making the test quick and easy, requiring only basic laboratory equipment such as a light microscope. Many of the large cohort studies assessing the utility of OxiSperm or the NBT assay in male infertility diagnosis have only assessed infertile men, with indicators for medium to high levels of ROS present in 31–76% of infertile participants [53,54,58,59]. When comparing OxiSperm^®^ to a wide range of clinical sperm DNA damage markers (including, TUNEL, Comet, SCSA and chromatin dispersion assay) in 50 fertile and 50 infertile men, Javod et al. found that OxiSperm^®^ had one of the highest levels of sensitivity, at 0.991, however, it had the lowest levels of specificity, at 0.322 and did not correlate with male infertility. In a study assessing 132 couples using donor oocytes, Pujol et al. found no association between OxiSperm reaction in semen and sperm parameters, fertility rates, embryo morphology nor pregnancy or live birth rates [58]. In a subset of 143 out of 707 included participant samples, Gosálvez et al. found that while the level of OxiSperm reaction in the neat semen were not associated with levels of sperm DNA damage as measured by HaloSperm (Madrid, Spain) (chromatin dispersion assay), it was predictive of the likelihood of sperm developing DNA damage after prolonged exposure to seminal plasma (up to 24 h at 37 °C) [54]. One major limitation of the NBT OxiSperm assay is that seminal plasma exhibit high levels of reductase which is capable of reducing the NBT to formazan creating false positive results in neat semen assays [63]. This may be contributing to its low predictive value. This limitation has somewhat been rectified in the newer OxiSperm^®^II assay which measures NBT reactivity in whole semen, seminal plasma and sperm within the same sample. This assay utilizes a gel that displays varying intensities once reacted that can be categories as low, medium or high as per the color scale. However, the semi-qualitative measurement of OxiSperm^®^II creates considerable opportunity for introducing bias into results while also showing low assay precision, as differences in color perception between scientists could result in different interpretations. While the self-contained OxiSperm kits are laboratory friendly and time effective, their lack of correlation with male infertility, fertilization rates or pregnancy outcomes, together with a lack of clinical trial data and low precision have likely contributed to their relatively low clinical uptake. This may partially explain the absence of OxiSperm kits from inclusion in the latest 6th edition WHO guidelines under the recommended assessments of ROS.

## 4. ROS Research Tools

The use of commercial fluorescent probes (i.e., ThermoFisher [Waltham, MA, USA]: ‘Oxidative Stress Detection’) for determining ROS in sperm are highly used and praised in laboratory settings [64], for their abilities to not only measure total intracellular ROS, but also specific ROS radicals that are often associated with pathology (i.e., OH^−^). An important factor limiting their potential to be used clinically is the requirement for a flow cytometer to perform the multiple probe analysis. A lack of large cohort studies validating the assays for male pre-conception care has also limited their capacity to be used beyond the research laboratory. The use of ROS research probes can also help determine whether specific radicals (Figure 2) are associated with male infertility, and thus could help to determine which ROS radicals should be measured clinically. As sperm require an endogenous level of ROS for normal sperm function, and with ‘optimal’ concentrations having not yet been fully determined, one may suggest that a better approach might be to instead measure downstream effects of oxidative stress on sperm, such as oxidative DNA damage or lipid peroxidation, given they have been shown to directly inhibit sperm function and fertilization [17,23]. Below, we outline the strengths and limitations of measuring these downstream markers, being oxidative DNA damage and lipid peroxidation, as opposed to specific ROS radicals.

### 4.1. Oxidative Sperm DNA Damage

8-hydroxy-2′-deoxyguanosine (8OHdG) is a DNA base adduct, commonly used as a biomarker of oxidative stress due to its association with nuclear and mitochondrial DNA damage [65]. Formation of 8OHdG in semen samples has previously been correlated with increases in DNA fragmentation, chromatin retention and decreases in sperm motility and fertilization rates, thus making it a promising potential biomarker for oxidative stress damage and male infertility [66]. Currently, 8OHdG can be quantified in semen samples using light microscopy, fluorescence microscopy and flow cytometry [65,67]. Major limitations arise in the difficult and often time-consuming protocols required for microscopy and flow cytometry techniques, with large and often expensive equipment also being required. While these assays have been validated as a reliable and accurate means of quantifying 8OHdG in semen, their application as a clinical diagnostic is severely limited and somewhat impractical. A now discontinued assay, OxyDNA Test^®^, used fluorescence to detect intracellular concentrations of 8OHdG, however, it had exceedingly low specificity and sensitivity, and no correlation to sperm quality of fertility potential [68]. Further, its low accuracy at detecting 8OHdG hindered its use in both clinical and research settings [69]. One study, comparing several ways of measuring 8OHdG concluded that assays utilizing antibody-mediated flow cytometry was the most accurate and reliable quantification method for determining sperm 8OHdG concentration [69]. The assay showed low intra- and inter- assay coefficients of variation and a cut-off of 65.8% for 8OHdG positive sperm. Another study reported similar findings, observing significant correlations between sperm DNA damage and 8OHdG formation using antibody mediated flow cytometry [66]. While the current literature appears supportive of the use of 8OHdG quantification in the assessment of male factor infertility, the requirement for a flow cytometer hinders its use in clinical settings. Hence, further attempts should be made to produce a novel diagnostic kit that allows for the quick, easy and most importantly, accurate measurements of 8OHdG positive sperm.

### 4.2. Lipid Peroxidation

Lipid peroxidation is one of the major consequences of oxidative stress and high concentrations of seminal ROS. These lipid peroxidation cascades result in the formation and accumulation of lipid aldehydes, including 4-hydroxy-2-nonenal (4HNE) and malondialdehyde (MDA) that are capable of disrupting sperm function through the formation of adducts with key proteins and DNA [70]. Lipid peroxidation can also be easily quantified in sperm through various commercially available stains.

### 4.3. BODIPY^TM^ 581/591-C11

The BODIPY-C11 is an extremely useful probe for determining peroxidative damage in human sperm as oxidation of the polyunsaturated butadienyl portion of the dye results in a shift of the fluorescence from ~590 nm (red) to ~510 nm (green) in live cells [71]. High fluorescence, detected by BODYIP-C11 has been negatively associated with sperm motility and positively correlated with ROS generation and genital urinal tract infection [71,72,73]. However, recently it was reported through mass spectrometry of oxidative products that BODPIY-C11 is more sensitive to oxidation than endogenous lipids, and as a result was found to under report the antioxidant effect of alpha-tocopherol and therefore, may be over-representing lipid peroxidation of a cell [74].

### 4.4. Thiobarbiuric Acid-Malondialdehyde (TBA-MDA)

TBA-MDA works by measuring the reaction of thiobarbiuric acid (TBA) and malondialdehyde (MDA), which occur during lipid peroxidation [75]. Quantification of lipid peroxidation products, via this assay, have been previously shown to correlate with reductions in sperm motility, morphology and count, thus deeming the assay reliable and accurate in assessing male infertility [76]. The time effective nature of the assay has seen it receive praise for being highly sensitive and accurate by some users, however others have questioned its clinical use as a results of its overestimation of MDA levels [77]. Whilst Garcia and colleagues [77] were able to establish effective methods to minimize this overestimation, inconsistencies in laboratory and clinical protocols, as well as equipment requirements, could create disparities in study design and data outputs from the assay.

### 4.5. 4-Hydroxy-2-Nonenal (4HNE)

Lipid peroxidation results in the formation and accumulation of cytotoxic break down products, including 4-hydroxynonenal (4HNE). This highly reactive lipid aldehyde can react with proteins and nucleic acids, as well as other lipids present in spermatozoa, impacting sperm function and fertilization potential by modifying germline proteins, altering protein homeostasis and cell function [78,79]. 4HNE can be quantified in semen samples using antibodies that detect and attach to the lipids which are then quantified using ELISA or visualized using either western blot or immunohistochemistry techniques. Previous literature has shown increased concentrations of 4HNE in human sperm displaying compromised motility, impaired membrane integrity and decreased oocyte-binding abilities [79,80,81]. Moreover, 4HNE has been shown to cause a self-perpetuating cycle stimulating increases in ROS produced by the mitochondria, further increasing lipid peroxidation [78]. Whilst quantifying 4HNE in sperm holds some promise in infertility diagnostics, the requirement for highly specialized and expensive equipment limits its implementation clinically. Further, immunoblots have limited sensitivity, and thus the low concentrations of 4HNE present in sperm may not always be accurately measured using these assays. There are also multiple 4HNE antibodies and protocols that are often used interchangeably between research laboratories, creating major limitations when comparing and analyzing data from multiple sources. However, a 2013 study comparing two 4HNE antibodies (commercial vs noncommercial) found significant correlations between them, with the authors concluding that both antibodies produced reliable and precise data [82]. The inter-assay and inter-day variance using both ELISA’s was minimal, however, the absolute values reported differed significantly between the two, with the most commonly commercially available antibody also recognizing additional amino acid residues alongside 4HNE [82]. This acts to reduce the accuracy and reliability of the commercial antibodies for quantifying 4HDE in samples, thereby potentially limiting its use as a diagnostic tool in clinics. Therefore, further attempts should be made to develop a stable, highly specific probe for the detection of lipid peroxidation in sperm.

## 5. Why We Are Still Likely Some Time Away from Semen ROS Measures Becoming a Mainstream Preconception Diagnostic Test in Men

### 5.1. Indiscrepancies in Stated Normal Reference Ranges

One of the biggest limitations of current clinical diagnostics of ROS in semen lies in the inconstancies and discrepancies between studies in what constitutes a normal concentration of ROS. Normal reference ranges are usually established by selecting a group of ‘healthy’ subjects to evaluate, however, the definition of ‘healthy’ in infertility research is usually poorly defined. In the cases of the studies presented in Table 2, the majority of the studies determined ‘reference ranges’ by comparing fertile and infertile men. This is because the definitions of ‘infertile men’ tends differ between studies, with some defining it as the inability to conceive spontaneously, others defining it as the need for ART and others through detecting abnormalities in the standard semen analysis. The same can be said for the definition of a ‘fertile man’. This of course, results in different clinical cut offs between studies as the populations assessed are not directly comparable.

Both analytical (accuracy and preciseness) and clinical factors (correct interpretation of male infertility) need to be considered when evaluating/validating the usefulness of a new medical device for the diagnosis of male infertility. Usually, a diagnostic assay validation dataset generally requires up to 640 data points for clinical use a Class II medical device or up to 2160 data points for premarket approval for clinical usages as a Class III medical device (C-path, Tucson, AZ, USA, June 2017) [83]. So far, MiOXSYS^TM^ is the only ROS medical device that has reported measurements in over 2000 men from multiple clinics [49]. While a large number of men were assessed, an important limitation of this study was that participants were restricted to men seeking fertility treatment. In standard biomarker validation and determination of normal reference ranges, samples are largely collected from the general population, as this often helps determine whether the fluctuations in assay ranges are due to individual variation, pathologies, commodities, treatments and environmental factors [83]. It is perhaps not surprising then that the latest WHO methods manual for the processing of human semen still advises caution for the use of both Luminol and MiOXSYS for the diagnosis of male infertility. Further large-scale multi-centered clinical trials are required for determining the clinical usefulness of these devices for the classification of male subfertility and infertility, with these studies also needing to extend their outcomes to include pregnancy and live births. This is extremely important because if these measures do not correlate with pregnancy or live birth outcomes, they will provide no additional clinical benefit in the diagnosis of male infertility over what is already provided by a standard semen analysis.

### 5.2. The Best Potential Diagnostic Tool to Measure ROS Might Be Dependent on Conception Method

An important consideration when developing new methods for measuring ROS in semen is that any specific test employed in a clinical ART setting may well differ from that used to measure ROS as part of routine male preconception care. This is largely due to differences in the individual components of the ejaculate that the gametes encounter during ART, compared to the female reproductive tissues during natural conception.

During ART cycles, including intrauterine insemination (IUI), standard in vitro fertilization (IVF) and intracytoplasmic sperm injection (ICSI), high quality motile sperm isolated following swim-up and sperm washing procedures are mostly stripped of the non-cellular portion of the ejaculate, referred to as seminal plasma. Importantly, seminal plasma provides a rich source of antioxidants which act to protect sperm from additional oxidative damage following ejaculation and its removal leaves the male gametes vulnerable to further oxidative or inflammatory damage [84,85]. It is essential therefore to consider whether ROS concentrations should be evaluated in sperm, seminal plasma, whole semen or perhaps only on the motile sperm population. In this setting, we might reasonably rationalize that understanding ROS concentrations in the motile sperm fraction may be most clinically informative in these patients, given the isolated sperm being tested are the very ones that will be used to fertilize the oocyte and participate in early embryonic events.

In contrast, during natural conception, the female reproductive tissues are exposed to the full ejaculate comprising sperm and seminal plasma. Traditionally, seminal plasma was believed to serve one main function, facilitating the movement of sperm into the female reproductive tract to facilitate oocyte fertilization [86]. However, there is now increasing evidence supporting a more complex role, with seminal plasma directly promoting fertility and fecundity through its effects on the female partner’s immune response and her reproductive processes [87]. Seminal plasma contains an abundance of cytokines, chemokines, and growth factors which are contributed from various sites within the male reproductive tract including leucocytes, Sertoli and Leydig cells in the testis [88,89], as well as the secondary male accessory glands comprising the epididymis, prostate, and seminal vesicles [90,91,92], some of which act to support sperm function [93,94] while others mediate the female response to seminal fluid [95,96]. The ready detection of cytokines in seminal plasma has made them an attractive target to evaluate their potential association with male fertility status [93,97,98] and more recently, seminal ROS concentration [99]. Indeed, several studies have reported a positive relationship between the presence of leucocytes in semen and ROS production [100], while the latter has also positively correlated with the abundance of seminal plasma pro-inflammatory cytokines such as C-X-C motif chemokine ligand 8 (CXCL8) [99,101], interleukin 6 (IL6) [99,102] and tumor necrosis factor (TNF) [99,103]. CXCL8 has also been reported as being elevated in the seminal plasma of men with genitourinary tract infection [93,104] where it is believed associated with reduced fertility [105,106]. Further, the 6th edition of WHO guidelines suggest quantifying the abundance of inflammatory mediators in seminal plasma may be a useful tool for Andrologist’s to support leucocyte counts in men with suspected genital tract infection [12]. While CXCL8 is just one example, measuring the abundance of specific seminal plasma cytokines may provide important insight into the likely source of ROS in individual men. Therefore, given the important contribution both sperm and seminal plasma play in preparing the female reproductive tissues for pregnancy during natural conception, careful consideration must be given as to whether measuring ROS in whole semen, sperm, seminal plasma, or a combination thereof will provide greatest diagnostic utility for evaluating male preconception health.

### 5.3. Fundemental Knowledge of the Natural Fluctuations in Men and the Most Influential Biological and Lifestyle Drivers for Inducing Oxidative Stress

It is easy to forget that sperm require a physiologically balanced level of ROS to perform normal biological functions [107,108,109], yet what this level is in humans, is still to be determined. Despite a number of studies modulating ROS concentrations in human sperm and showing detrimental effects to basic measurements [27,71,110,111], more advanced assessments of sperm function, such as sperm binding, are lacking. Increased sperm ROS concentrations in animal models, induced through exposure to H_2_O_2,_ have been reported to cause a delay in time to first mitosis and embryonic on-time development (8-cell and blastocyst) and a decrease in blastocyst total and inner cell mass cell numbers, leading to reduced implantation rates following embryo transfer [9,112,113]. It is of upmost importance that we understand what the ‘goldilocks’ zone is for ROS concentrations in sperm necessary for optimal fertilization and embryo viability, as it is possible that this is not linear. Therefore, without a basic understanding of what ‘normal’ semen ROS concentrations are and how they fluctuate over time in men, it is not possible to know what level of semen/sperm ROS we are wanting men or the clinic to achieve prior to conception.

Very few studies have assessed the fluctuations in semen ROS overtime. Using luminol chemiluminescence Zorn et al. found that ROS levels were consistent in 25 infertile men who had two measures over a six month period [114]. Using MiOXSYS^TM^, Agarwal et al., found that ORP fluctuated based on semen analysis in 28 fertile men who had repeat semen samples after 3–5 month follow up, with ORP decreasing as sperm concentration and motility increased [46]. However, this is not surprising given that normed ORP is divided by sperm count. A longitudinal study assessing repeat semen samples from a healthy proven fertile individual over 21 months showed significant variations in ROS (chemiluminescence) over time that were independent of basic sperm measurements [115]. They concluded that the differences were likely due to changes in ejaculation frequency and seasonal/lifestyle variations [115]. While we know that many biological and lifestyle factors are able to influence the production of ROS in semen [23,116], we lack fundamental knowledge in terms of the relative contributions of each of these biological and lifestyle factors to oxidative stress in sperm. This is further highlighted in a recent report from the Australian peak body for male reproductive health, Healthy Male, entitled ‘Paternal Plus- A Case for Change’ [3], which reported that there was a lack of good quality evidence to inform preconception health advice for men, despite health professionals wanting to seek education and information to support their engagement of men during the preconception period. For instance, if a man diagnosed with high semen ROS smokes cigarettes, drinks alcohol daily, works in a factory and has a calorie dense nutrient low diet, which factor/s should they be told to modify first? It is highly unlikely they would be able to modify all factors at once and it might be found that simply changing one factor might significantly improve outcomes. For example, the addition of a handful of mixed nuts daily to 119 healthy men who consumed a western diet was sufficient to reduce sperm DNA damage [117]. However, without large cohort studies assessing sperm ROS concentrations in men throughout aging and studying a wide variety of lifestyle and biological factors will we be unable to determine which factors we should be targeting first for intervention.

### 5.4. A Better Understanding of the Usefulness of In Vivo and In Vitro Antioxidant Interventions for Restoring Sperm Oxidative Injruy

The use of oral antioxidants for reducing the effects of oxidative stress in men experiencing infertility is widely practiced, given the positive association between dietary intake and both circulating blood and seminal plasma concentrations [118,119]. Antioxidants inhibit or delay the oxidation of molecules ether through scavenging or by chelation of redox radicals [120]. The most widely studied oral antioxidants for the treatment of male infertility include vitamins E, B and C, carotenoids, carnitines, coenzyme Q10, cysteine and the micronutrients selenium, zinc and folate [121]. Most are given as either single or combined daily supplements and can often be offered as a first-round treatment for couples experiencing infertility, given their ready availability and low cost [122]. Despite, their widespread use in the treatment of male infertility, the evidence supporting their beneficial effect remains quite poor. A systematic review published in 2019 in the Cochrane Database of Systematic Reviews found that the evidence for the use of oral antioxidants to improve pregnancy rates and live births in men with infertility was of ‘low’ to ‘very low’ quality, with the biggest limitations of current trials been inadequate pregnancy and live birth outcomes reported (only 12/44 included studies) [123]. While, it appears as though oral antioxidants may lead to increased live birth rates (OR 1.79, 95% CI 1.20–2.67), when studies at high risk of bias were removed from the analysis, the evidence for increased live birth was lost (OR 1.38, 95% CI 0.89–2.16) [123]. The authors concluded that sub fertile couples should be advised that overall, the evidence for the use of oral antioxidants for the treatment of male infertility is inconclusive based on serious risk of study bias due to poor reporting of methods of randomization, often unclear or high attrition, small over all samples size and limited studies assessing pregnancy and live birth outcomes [123]. These results were similarly mirrored in a systematic review written by Agarwal et al. [122] in 2021, who found that while antioxidant supplementation improved semen quality in infertile men, there was little evidence supporting a beneficial effect of supplementation on live birth rates. Therefore, larger well-designed randomized placebo-controlled trials with the primary outcome of assessing live birth rates are required to determine the utility of oral antioxidant therapy for the treatment of male infertility. Further, understanding the best types, combination and dosage of oral antioxidants for improving sperm function and pregnancy outcomes is also lacking despite a number of male preconception supplements already being marketed to men (i.e., Fertility Smart, Conception Men, Menvit, NaturoBEST, ConceptionXR).

Oral antioxidants are not the only way researchers have been trying to reduce sperm oxidative stress levels. Several studies have now investigated whether the addition of antioxidants to sperm preparation media prior to IVF treatment is beneficial [124,125,126,127,128]. As mentioned earlier, in standard IVF sperm are stripped of their seminal plasma which contains some of the most highly specialized forms of antioxidants and scavenging enzymes known including; glutathione peroxidase (GPx5), extracellular superoxide dismutase (SOD), uric acid, vitamin C, tyrosine and polyphenols [129,130]. The process of sperm washing therefore, completely removes all the sperm extrinsic antioxidant components, rendering sperm defenseless to oxidative damage. Then, since sperm are proficient generators of ROS they are left with only their own poorly functioning antioxidant defenses (SOD, GPX and catalase) to combat oxidative damage [110,131]. These studies have found that the addition of EDTA, lycopene, zinc, ascorbic acid (vitamin C), coenzyme Q10, taurine and glutathione to the sperm preparation medium was able to increase sperm motility and viability and decrease sperm ROS production and lipid peroxidation [124,125,126,127,128]. A limiting factor of current studies is the addition of antioxidants that are not cell permeable [132], therefore, the positive effects of supplementation seen in these studies maybe more due to extrinsic ROS removal via antioxidant scavenging, as opposed to reducing sperm ROS generation and intrinsic ROS scavenging. Similar to oral antioxidant intake studies, only a few in vitro studies have examined the downstream effects on fertilization, embryo development and live birth reporting favorable outcomes [133,134,135,136,137], which are required in order to provide an evidence base prescribing use of antioxidants in clinical practice. This is important as excessive amounts of antioxidants can have the opposite effect, interfering with physiological ROS concentrations, leading to enhanced ROS generation in mitochondria and lead to further oxidative injury to cells [138,139]. Therefore, the concentration of antioxidants added to sperm preparation media needs to be carefully and thoroughly researched, as a single concentration may not be suitable for all men, especially if they have a “balanced or normal level” of ROS which could have the undesired effect of worsening outcomes. This worsening of outcomes was precisely what we observed in preliminary experiments from our own laboratory (Figure 3). Utilizing a mouse model of sperm H_2_O_2_ exposure (3000 µM) prior to standard insemination (IVF) we added 100 µM of manganese (III) 5,10,15,20-tetrakis(4-carboxyphenyl)porphyrin-21,23-diide chloride (MnTBAP), a cell-permeable superoxide dismutase (SOD) mimetic and peroxynitrite scavenger [140] post H_2_O_2_ exposure (Figure 3). While this concentration was able to reduce sperm mitochondrial superoxide concentrations by more than 50% (Figure 3b), which is consistent with the data from human studies using similar concentrations [141,142], surprisingly, fertilization rates were significantly reduced, with a ~50% reduction in 2-cell cleavage rates 24 h post insemination with MnTBAP treated sperm (Figure 3c,d). This occurred even after the MnTBAP was washed out prior to insemination and a trend for increased sperm motility was seen (*p* = 0.08, Figure 3a). This serves as an important reminder that while the addition of antioxidants to sperm culture media may be able to reduce sperm ROS concentrations and therefore, appear promising as a potential treatment for male factor infertility, we must first thoroughly understand the mechanism of action for each of the antioxidants and the downstream consequences of using them on fertilization, embryo development and pregnancy outcomes, before supporting their addition to sperm preparation media.

## 6. Conclusions

The evaluation of ROS in semen shows some promise as a diagnostic tool for male infertility and potentially in determining the likelihood of having a healthy pregnancy and child. While several clinically available tests show promise for clinical Andrology use, discrepancies in their documented reference ranges and the lack of cohort’s studies/clinical trials assessing the benefit of their use on fertilization rates, embryo development, pregnancy and live birth rates will continue to limit their clinical usefulness outside that of a standard semen analysis. Further, we do not fully understand some of the fundamentals around redox biology in sperm including (1) what are the optimal ROS concentrations for normal sperm function, successful fertilization and a healthy child? (2) What are the natural variations/fluctuations in ROS concentrations between and within men over time? (3) What are the most influential lifestyle and biological drivers? (4) Are there any benefits to either in vivo or in vitro antioxidant exposure prior to conception, and (5) Is the best ROS diagnostic tool dependent on conception method? Until these questions are answered and we are better positioned to understand which men would benefit most from this testing, what to actually measure and what the best interventions and therapeutic treatment options available might be, the use of clinical ROS detection methods will likely continue to be used by only a handful of clinics worldwide.

## Figures and Tables

**Figure 1 antioxidants-11-00264-f001:**
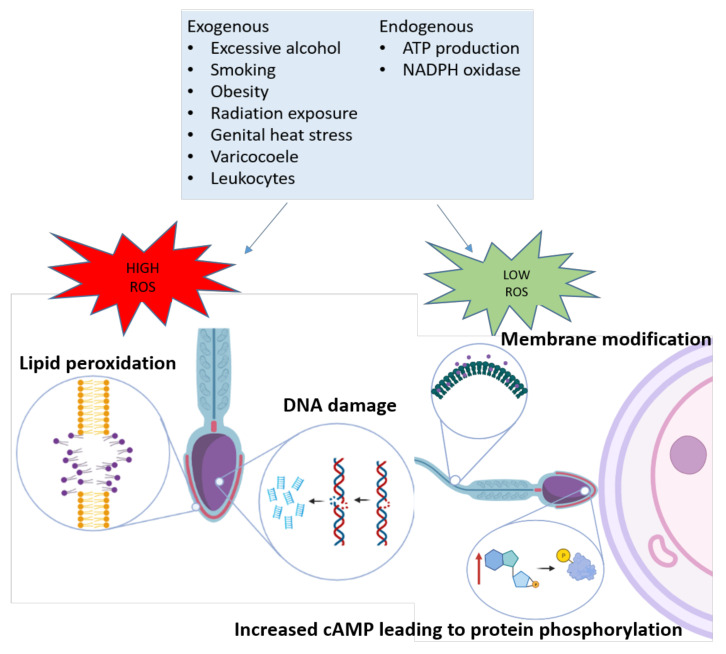
Sources of ROS generated exogenously and endogenously in sperm. At low levels, ROS contributes to successful fertilization by increasing membrane fluidity through cholesterol exudation and through tyrosine phosphorylation of target proteins required for fertilization. Certain ROS species directly inhibit tyrosine phosphatases and others contribute to the activation of cyclic adenosine monophosphate (cAMP), leading to protein kinase A activation and phosphorylation of target proteins.

**Figure 2 antioxidants-11-00264-f002:**
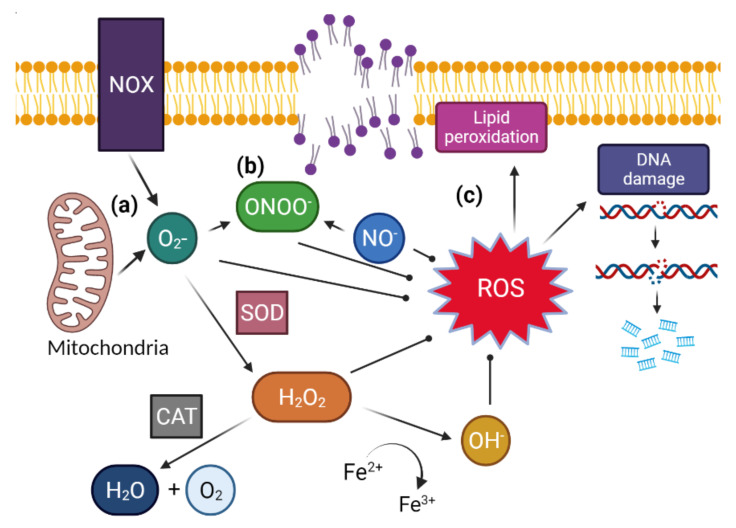
(**a**) ROS formed within the cell starts as superoxide (O_2_^−^) created from ATP generation in mitochondria and from NADPH oxidase (NOX), and superoxide dismutates, facilitated by the protein superoxide dismutase (SOD) into hydrogen peroxide (H_2_O_2_). Hydrogen peroxide can then be neutralized by catalase (CAT) to create water (H_2_O) and singlet oxygen (O_2_). However, in the presence of ferrous iron it will react, creating hydroxyl radical (OH^−^). (**b**) If there is nitric oxide (NO^−^) present, then it will react to create peroxynitrite (ONOO^−^), a potent but unstable oxidant. (**c**) These ROS contribute to sperm damage through oxidation of the guanine base in DNA, creating single strand breaks, double strand breaks and DNA fragmentation, or through its oxidation of polyunsaturated fatty acids, creating lipid radicals that self-propagate, resulting lipid peroxidation.

**Figure 3 antioxidants-11-00264-f003:**
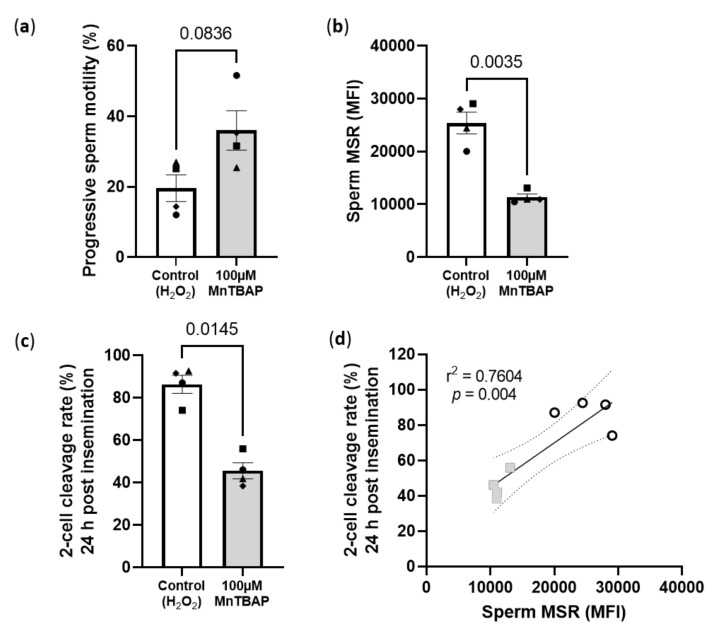
Reducing sperm mitochondrial superoxide concentrations with 100 µM MnTBAP in a mouse model of in vitro H_2_O_2_ exposure significantly impairs 2-cell embryo development. (**a**) Proportion of sperm with progressive motility; (**b**) Sperm mitochondrial superoxide mean fluorescence intensity (MFI) (MitoSox Red, (MSR) a specific mitochondrial superoxide indicator); (**c**) Proportion of 2-cells 24 h post insemination with 10,000 sperm and (**d**) Linear regression of sperm superoxide concentrations and 2-cell cleavage rates (grey squares: 100 µM MnTBAP, white circles: 3000 µM H_2_O_2_). Data is representative of 4 CBAF1 males (represented by different symbols on graphs) and 24 super ovulated 3–4 weeks old CBAF1 females. Female mice were super ovulated with 5IU of PMSG followed 48 h with 5IU of HCG. Sperm was collected from male mice 15 h post HCG and incubated in 3000 µM of H_2_O_2_ in G-IVF PLUS (Vitrolife) for 1 h at 37 °C, 5% O_2_ and 6% CO_2_. Sperm samples were split into two groups and either incubated in (1) Control: 3000 µM of H_2_O_2_ in G-IVF PLUS or (2) 100 µM of MnTBAP: 3000 µM of H_2_O_2_ in G-IVF PLUS for a further 1 h at 37 °C, 5% O_2_ and 6% CO_2_. Cumulus oocyte complexes (COC) were collected from female mice 15.5 h post HCG. Sperm were washed of their treatments after backfilling with 3 mL of G-IVF PLUS by centrifugation 400× *g* for 5 min. COCs were inseminated with 10,000 sperm 16.5 h post HCG. Following a 4 h fertilization, zygotes were washed and cultured in G1 PLUS (Vitrolife) for 24 h at 37 °C, 5% O_2_ and 6% CO_2_ at which 2-cell embryo cleavage rates were assessed. Prior to insemination, sperm progressive motility was assessed in 200 sperm per sample and superoxide concentrations assessed after 30 min incubation at 37 °C of 5µM of MSR and 10,000 sperm assessed on a FACS Canto II. Data was analysed by a paired t-test or a simple linear regression showing the slope and 95% confidence intervals.

**Table 1 antioxidants-11-00264-t001:** Strengths and limitations of current commercially available ROS detection tools for male infertility.

Assay	Company	What it Measures	Strengths	Limitations
**Luminol Chemiluminescence**	Multiple companies 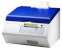	Luminol is first oxidized by many radicals (i.e., -OH and CO_3_) and peroxidases, forming the luminol radical. The luminol radical then reacts with superoxide, forming the short-lived intermediate hydroperoxide. Hydroperoxide is decomposed to 3-aminophyhalane, which emits light.	High quantum yieldHighly sensitiveEasy to measureReadily availableHighly reproduceble	Cannot differentiate different ROS radicalsLarge semen volume required (~400 µL).Cannot differentiate mature sperm from immature sperm or other cell types (i.e., leucocytes).No defined references ranges/decision limitsCannot be used on frozen samplesTemperature-sensitive >25 °CLight-sensitive and can oxidize over time
**MiOXSYS**	MiOXSYS 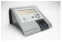	Measures the transfer of electrons from oxidants to antioxidants (sORP)	Rapid result (~5 min)Highly sensitiveSmall sample volume required (30 µL)Good reproducibilityObtain accurate results up to 2 h post ejaculationCan be used on fresh or frozen samplesCost effective	No definitive reference ranges/decision limitsCannot differentiate different ROS radicalsTemperature-sensitive between 2 and 37 °C
**OxiSperm**	Halotech DNA 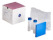	When exposed to superoxide, the NBT reagent is reduced and converted into blue formazan crystals.	Time-effectiveHighly sensitiveCost-effectiveRapid result (~15 min)Small sample volume required (~30–50 µL)	Subjective analysisLow assay precisionOnly measures one ROS radicalSeminal plasma reductase interferenceNot found to correlate with fertilization, embryo development or pregnancy outcomes in ART

ART: Assisted Reproductive Technology, Temp: Temperature.

**Table 2 antioxidants-11-00264-t002:** Relevant cohort studies assessing/validating Luminol, MiOXSYS and OxiSperm (NBT).

Reference	Population	Assay	Primary Finding
[31] Homa, S.T.; Vessey, W.; Perez-Miranda, A.; Riyait, T.; Agarwal, A. Reactive oxygen species (ros) in human semen: Determination of a reference range. *Journal of assisted reproduction and genetics* 2015, *32*, 757–764.	Men attending routine semen analysis; *n* = 94 normal semen parameters, *n* = 100 abnormal semen parameters but low leucocytes and *n* = 41 any semen parameters with leucocytospermia	Luminol	Significantly different between Groups 1, 2 and 3 19.75 ± 8.12, 95.03 ± 33.63 and 890.17 ± 310.23 RLU/sec/10^6^ sperm, respectively Cut off < 24.1 RLU/sec/10^6^ (specificity 87.2%, sensitivity 80.5%)
[32] Ochsendorf, F.R.; Thiele, J.; Fuchs, J.; Schüttau, H.; Freisleben, H.J.; Buslau, M.; Milbradt, R. Chemiluminescence in semen of infertile men. *Andrologia* 1994, *26*, 289–293.	*n* = 49 consecutive infertile men, and *n* = 20 controls	Luminol	ROS cut off 1.5 × 10^5^ counted photons per min^−1^/2 × 10^6^ sperm
[33] Athayde, K.S.; Cocuzza, M.; Agarwal, A.; Krajcir, N.; Lucon, A.M.; Srougi, M.; Hallak, J. Development of normal reference values for seminal reactive oxygen species and their correlation with leukocytes and semen parameters in a fertile population. *J. Androl.* 2007, *28*, 613–620.	*n* = 114 fertile men seeking vasectomy and *n* = 47 subfertile men	Luminol	Without leucocytes ROS cut off of 5 × 10^3^ counted photons per min/2 × 10^5^ sperm (Specificity 76.0%, sensitivity 73.3%) With leucocytes ROS cut off of 1.25 × 10^4^ counted photons per min/2 × 10^5^ sperm (Specificity 66.7%, sensitivity 71.9%)
[34] Fingerova, H.; Oborna, I.; Novotny, J.; Svobodova, M.; Brezinova, J.; Radova, L. The measurement of reactive oxygen species in human neat semen and in suspended spermatozoa: A comparison. *Reproductive Biology and Endocrinology* 2009, *7*, 118.	*n* = 91 infertile men and *n* = 34 men with proven fertility	Luminol	ROS = 0.26 RLU/10^3^ proven fertile vs. 1.1 RLU/10^3^ for semen abnormalities
[35] Desai, N.; Sharma, R.; Makker, K.; Sabanegh, E.; Agarwal, A. Physiologic and pathologic levels of reactive oxygen species in neat semen of infertile men. *Fertil. Steril.* 2009, *92*, 1626–1631.	*n* = 54 infertile men and *n* = 51 fertile donors	Luminol	ROS ≥ 1.85 × 10 counted photons per min/2 × 10^5^ sperm highly predictive of infertility (77.8% sensitivity and 82.4% specificity)
[36] Venkatesh, S.; Shamsi, M.B.; Dudeja, S.; Kumar, R.; Dada, R. Reactive oxygen species measurement in neat and washed semen: Comparative analysis and its significance in male infertility assessment. *Archives of gynecology and obstetrics* 2011, *283*, 121–126.	*n* = 65 infertile men with abnormal semen parameters, *n* = 17 infertile with normal semen parameters and *n* = 43 fertile controls	Luminol	ROS = 3.44 × 10^4^ RLU/min/20 million sperms for men with abnormal semen parameters vs. 7.9 × 10^3^ RLU/min/20 million for infertile normal semen and 3 × 10^2^ RLU/min/20 million for fertile controls
[37] Agarwal, A.; Tvrda, E.; Sharma, R. Relationship amongst teratozoospermia, seminal oxidative stress and male infertility. *Reprod. Biol. Endocrinol.* 2014, *12*, 45.	*n* = 79 tetratozoospermic men and *n* = 56 healthy donors	Luminol	ROS cut off of 93 RLU/sec/10^6^ sperm (specificity of 70.4% and sensitivity of 61.4%)
[38] Novotny, J.; Aziz, N.; Rybar, R.; Brezinova, J.; Kopecka, V.; Filipcikova, R.; Reruchova, M.; Oborna, I. Relationship between reactive oxygen species production in human semen and sperm DNA damage assessed by sperm chromatin structure assay. *Biomedical papers* 2013, *157*, 383–386.	*n* = 39 men from infertile couples and *n* = 23 fertile men	Luminol	Control group = 2.92 (2.32, 3.60), normospermia = 3.78 (3.09, 4.40) and semen abnormality = 4.02 (3.79, 4.29) log RLU/min/2 × 10^5^ sperm
[39] Agarwal, A.; Ahmad, G.; Sharma, R. Reference values of reactive oxygen species in seminal ejaculates using chemiluminescence assay. *J. Assist. Reprod. Genet.* 2015, *32*, 1721–1729.	*n* = 92 controls and *n* = 258 infertile men	Luminol	ROS cut off of 102.2 RLU/sec/10^6^ sperm (sensitivity 76.4% and specificity 53.3%)
[40] Agarwal, A.; Sharma, R.K.; Sharma, R.; Assidi, M.; Abuzenadah, A.M.; Alshahrani, S.; Durairajanayagam, D.; Sabanegh, E. Characterizing semen parameters and their association with reactive oxygen species in infertile men. *Reprod. Biol. Endocrinol.* 2014, *12*, 33.	*n* = 56 fertile donors and *n* = 318 infertile men	Luminol	ROS cut off of 91.9 RLU/sec/10^6^ sperm (sensitivity 93.8%, specificity 68.8%)
[41] Vessey, W.; Perez-Miranda, A.; Macfarquhar, R.; Agarwal, A.; Homa, S. Reactive oxygen species in human semen: Validation and qualification of a chemiluminescence assay. *Fertility and sterility* 2014, *102*, 1576–1583.e1574.	*n* = 23 semen samples from 19 men attending semen analysis	Luminol	No significant intra-or inter assay variation Working reagents stable for 3 months ROS measurements in samples are not stable and decline immediately after ejaculation
[42] Zorn, B.; Vidmar, G.; Meden-Vrtovec, H. Seminal reactive oxygen species as predictors of fertilization, embryo quality and pregnancy rates after conventional in vitro fertilization and intracytoplasmic sperm injection. *Int. J. Androl.* 2003, *26*, 279–285.	*n* = 147 male partners of infertile couples (41 IVF and 106 ICSI)	Luminol	High ROS classified as 10 mV/sec/10^9^ sperm observed in 43% of men Log ROS negatively correlated with fertilization above 25%; however, this was lost after adjusting for female and cycle characteristics Log ROS negatively correlated with embryo morphology after day 4 after multiple regression analysis Negative effect of ROS on pregnancy rates after IVF but not with ICSI
[43] Majzoub, A.; Arafa, M.; Mahdi, M.; Agarwal, A.; Al Said, S.; Al-Emadi, I.; El Ansari, W.; Alattar, A.; Al Rumaihi, K.; Elbardisi, H. Oxidation-reduction potential and sperm DNA fragmentation, and their associations with sperm morphological anomalies amongst fertile and infertile men. *Arab journal of urology* 2018, *16*, 87–95.	*n* = 1168 infertile men and *n* = 100 fertile from general population and infertility clinics	MiOXSYS	ORP = 1.73 mV/10^6^/mL (sensitivity 76% and 56% specificity)
[44] Agarwal, A.; Roychoudhury, S.; Sharma, R.; Gupta, S.; Majzoub, A.; Sabanegh, E. Diagnostic application of oxidation-reduction potential assay for measurement of oxidative stress: Clinical utility in male factor infertility. *Reproductive biomedicine online* 2017, *34*, 48–57.	*n* = 106 infertile men and *n* = 51 fertile men	MiOXSYS	Cut-off value of 1.39 mV/10^6^/mL (sensitivity 69.6% and specificity 83.1%)
[45] Agarwal, A.; Sharma, R.; Roychoudhury, S.; Du Plessis, S.; Sabanegh, E. Mioxsys: A novel method of measuring oxidation reduction potential in semen and seminal plasma. *Fertil. Steril.* 2016, *106*, 566–573.e10.	*n* = 33 infertile men and *n* = 26 fertile men from the general population	MiOXSYS	Cut-off value of 1.48 mV/10^6^/mL in semen (sensitivity 60% and specificity 75%) and 2.09 mV in seminal plasma (sensitivity 46.7% and specificity 81.8%)
[46] Agarwal, A.; Wang, S.M. Clinical relevance of oxidation-reduction potential in the evaluation of male infertility. *Urology* 2017, *104*, 84–89.	*n* = 194 infertile men and *n* = 29 men with repeat semen analysis	MiOXSYS	Cut-off value of 1.57 mV/10^6^/mL to detect one semen defect (sensitivity 70.4%, specificity 88.1) Cut-off value of 2.59 mV/10^6^/mL for detecting oligozoospermia (sensitivity 88%, specificity 91.2%)
[47] Vassiliou, A.; Martin, C.H.; Homa, S.T.; Stone, J.; Dawkins, A.; Genkova, M.N.; Skyla Dela Roca, H.; Parikh, S.; Patel, J.; Yap, T.; et al. Redox potential in human semen: Validation and qualification of the miox(sys) assay. *Andrologia* 2021, *53*, e13938.	*n* = 286 men undergoing routine semen analysis, and *n*= 854 samples for luminol validation	MiOXSYS and Luminol	No relationship between luminol RLU sec/10^6^ and sORP mV/10^6^/mL. A number of samples classified as low for MiOXSYS (<1.34 mV/10^6^/mL) were classified as high ROS by luminol (cut off value of 13.8 RLU/sec/10^6^, 86% sensitivity and 86% specificity). MiOXSYS was reproducible across operators, analyzers and days.
[48] Agarwal, A.; Du Plessis. SS.; Sharma, R.; Samanta, L.; Harlev, A.; Ahmad, G.; Gupta, S & Sabanegh, ES. Establishing the oxidation-reduction potential in semen and seminal plasma. *Fertil Steril* 2015, *104*, e146.	*n* = 18 fertile men	MiOXSYS	Cut-off = 4.73 mV/10^6^/mL (sensitivity = 100%, specificity = 89.5%) in sperm and 4.65 mV/mL (sensitivity = 100%, specificity = 93.8%) in seminal plasma
[49] Agarwal, A.; Panner Selvam, M.K.; Arafa, M.; Okada, H.; Homa, S.; Killeen, A.; Balaban, B.; Saleh, R.; Armagan, A.; Roychoudhury, S., et al. Multi-center evaluation of oxidation-reduction potential by the mioxsys in males with abnormal semen. *Asian journal of andrology* 2019, *21*, 565–569.	*n* = 2092 men attending for semen analysis from 9 countries	MiOXSYS	Cut-off 1.34 mV/10^6^/mL (sensitivity 98.1% and specificity 40.6%)
[50] Panner Selvam, M.K.; Henkel, R.; Sharma, R.; Agarwal, A. Calibration of redox potential in sperm wash media and evaluation of oxidation-reduction potential values in various assisted reproductive technology culture media using mioxsys system. *Andrology* 2018, *6*, 293–300.	(i) ENHANCE WG (Vitrolife, San Diego, CA, USA); (ii) Quinn’s™ Sperm Washing Medium (SAGE, In-Vitro Fertilization, Inc., Trumbull, CT, USA); and (iii) one sperm cryopreservation medium (Freezing Medium; Test Yolk buffer, Irvine Scientific, CA, USA).	MiOXSYS	ORP (mV) in sperm prep media = 267.3 mV
[51] Panner Selvam, M.K.; Agarwal, A.; Henkel, R.; Finelli, R.; Robert, K.A.; Iovine, C.; Baskaran, S. The effect of oxidative and reductive stress on semen parameters and functions of physiologically normal human spermatozoa. *Free Radic. Biol. Med.* 2020, *152*, 375–385.	*n* = 66 fertile men	MiOXSYS	sORPmV > 1.48 mV/10^6^/mL or < 9.76 mV/10^6^
[52] Sallam, N.; Hegab, M.; Mohamed, F.; El-Kaffash, D. Effect of oxidative stress in semen, follicular fluid and embryo culture medium on the outcome of assisted reproduction. *AIMJ* 2017, *2*, 59–65.	*n* = 50 couples with unexplained infertility undergoing IVF and ICSI	MiOXSYS	Cut off sORPmV < 1.57 mV/10^6^/mL for fertilization and < 0.75 mV/10^6^/mL for clinical pregnancy
[53] Iommiello, V.M.; Albani, E.; Di Rosa, A.; Marras, A.; Menduni, F.; Morreale, G.; Levi, S.L.; Pisano, B.; Levi-Setti, P.E. Ejaculate oxidative stress is related with sperm DNA fragmentation and round cells. *Int. J. Endocrinol.* 2015, *2015*, 321901.	*n* = 56 infertile men	OxiSperm in relation to DFI	L3 or L4 of semen oxidative stress correlated with DFI ≥ 30%
[54] Gosálvez, J.; Coppola, L.; Fernández, J.L.; López-Fernández, C.; Góngora, A.; Faundez, R.; Kim, J.; Sayme, N.; de la Casa, M.; Santiso, R., et al. Multi-centre assessment of nitroblue tetrazolium reactivity in human semen as a potential marker of oxidative stress. *Reproductive biomedicine online* 2017, *34*, 513–521.	*n* = 707 infertile men	OxiSperm	76% participants categorised as L2 (medium), only 4% L3 (high) and 20% L1 (low)
[55] Tunc, O.; Thompson, J.; Tremellen, K. Development of the nbt assay as a marker of sperm oxidative stress. *International journal of andrology* 2010, *33*, 13–21.	*n* = 21 fertile and *n* = 36 infertile men	NBT-reactivity	Cut-off = 24 μg formazan/10^7^ sperm (sensitivity 91.7%, specificity 81.0%)
[56] Esfandiari, N.; Sharma, R.K.; Saleh, R.A.; Thomas, A.J., Jr.; Agarwal, A. Utility of the nitroblue tetrazolium reduction test for assessment of reactive oxygen species production by seminal leukocytes and spermatozoa. *Journal of andrology* 2003, *24*, 862–870.	*n* = 21 infertile men and *n* = 9 healthy donors	NBT-reactivity	NBT positive sperm increased in samples with leucocytes present. Cut-off = 19% (sensitivity of 100% and specificity 86.4%)
[57] Amarasekara, D.S.; Wijerathna, S.; Fernando, C.; Udagama, P.V. Cost-effective diagnosis of male oxidative stress using the nitroblue tetrazolium test: Useful application for the developing world. *Andrologia* 2014, *46*, 73–79.	*n* = 102 subfertile and *n* = 30 proven fertile men	NBT-reactivity	Cut-off = 42.02 μg formazan/10^7^ sperm (sensitivity 71.4% and specificity 70%)
[58] Pujol, A.; Obradors, A.; Esteo, E.; Costilla, B.; García, D.; Vernaeve, V.; Vassena, R. Oxidative stress level in fresh ejaculate is not related to semen parameters or to pregnancy rates in cycles with donor oocytes. *Journal of assisted reproduction and genetics* 2016, *33*, 529–534.	*n* = 132 infertile men	OxiSperm	43.2% were in high oxidative stress (L3) and 30.3% were low (L2) and 25.0% very low (L1) No association between oxidative stress and fertilization rate, embryo morphology or pregnancy rates
[59] Degirmenci, Y.; Demirdag, E.; Guler, I.; Yildiz, S.; Erdem, M.; Erdem, A. Impact of the sexual abstinence period on the production of seminal reactive oxygen species in patients undergoing intrauterine insemination: A randomized trial. *The journal of obstetrics and gynaecology research* 2020, *46*, 1133–1139.	*n* = 90 infertile men	OxiSperm	Increased pigment staining related to higher ROS levels in 70% of samples with ejaculation length >4 days vs. 50% for 3–4 days abstinence and 43.3% for 0–2 days abstinence.
[60] Javed, A.; Talkad, M.S.; Ramaiah, M.K. Evaluation of sperm DNA fragmentation using multiple methods: A comparison of their predictive power for male infertility. *Clinical and experimental reproductive medicine* 2019, *46*, 14–21.	*n* = 50 infertile and *n* = 50 fertile men	OxiSperm	Fertile group L1 (low), 39%; L2 (low–medium); 24%; L3 (medium), 11%; and L4 (high), 36%. Infertile group L1 (low), 16%; L2 (low-medium), 11%; L3 (medium), 31%; and L4 (high), 42%. No correlations with male infertility (specificity 0.32, sensitivity 0.99)

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
