# Peer review of "Measuring Reactive Oxygen Species in Semen for Male Preconception Care: A Scientist Perspective"

_antioxidants, 2022, doi:10.3390/antiox11020264_

Round 1

Reviewer 1 Report

Dear Authors:

The manuscript by Castleton et al has demonstrated the role of ROS in semen, which is excellent and well-organized, strongly suggest for publishing. I have just a few suggestions.

Please add more background information about reactive oxydative species. ROS also plays an important role in cancer development like breast cancer and ischemic stroke complications and so on.  (Please cite: 1. Chen et al. Semin Cancer Biol. 2020 Oct 6:S1044-579X(20)30203-0. doi: 10.1016/j.semcancer.2020.09.012.  2. Shekhar et al. International Journal of Molecular Sciences. 2021; 22(4):2074. https://doi.org/10.3390/ijms22042074)

Best,

Author Response

Reviewer 1

  • Please add more background information about reactive oxydative species. ROS also plays an important role in cancer development like breast cancer and ischemic stroke complications and so on.  (Please cite: 1. Chen et al. Semin Cancer Biol. 2020 Oct 6:S1044-579X(20)30203-0. doi: 10.1016/j.semcancer.2020.09.012.  2. Shekhar et al. International Journal of Molecular Sciences. 2021; 22(4):2074. https://doi.org/10.3390/ijms22042074)

Response: We thank the reviewer for their kind words regarding the manuscript. As suggested we have added in a lines containing more generalised information about ROS and how it plays important roles in the development of many disease with reference to the suggested articles.

‘An over production of ROS has been implicated in the development of aging and many chronic and degenerative diseases including cancer, chronic obstructive pulmonary disease, cardiovascular disease, neurodegenerative disorders (Alzheimer’s, Huntington’s Parkinson’s, amyotrophic lateral sclerosis, spinocerebellar ataxia, ischemic stroke) and inflammatory bowel disease [13-15].’

Reviewer 2 Report

In this review entitled „Measuring reactive oxygen species in semen for male preconception care: A scientist perspective.“ Castleton et al describe how ROS are measured and used in the diagnostic for men fertility and describe the potential but also the pitfalls for the use of the ROS levels in this diagnostic. The general outline of the manuscript is very precise and the review is well written. However they are some minor points to clarify:

  • In figure 1 the sources of ROS are outlined. Please specify in the figure title that it refers to the sources of ROS in semen as there are of course other endogenous sources for ROS such as Xanthine oxidase or uncoupled NO synthase. In addition whereas in the main text leukocytes are referred as endogenous sources in the figure 1 itself they are pointed out as exogenous sources. Please clarify. In addition, in the figure legend the activation of cAMP is mentioned, but not shown in the figure itself, please clarify this more in the figure.
  • Please check the consistent chemical formula for superoxide in the text and all figures. In figure 2 it is referred as “O2.-“ in the text it is “O2-“ Please make this consistent.
  • Please check the reference in the text to figure 3, as Figure 3B is not referred to in the text. In addition, MitoSox Red is a specific mitochondrial superoxide indicator. Therefore please specify this circumstance in the text and figure legend.

Author Response

Reviewer 2

  • In figure 1 the sources of ROS are outlined. Please specify in the figure title that it refers to the sources of ROS in semen as there are of course other endogenous sources for ROS such as Xanthine oxidase or uncoupled NO synthase. In addition whereas in the main text leukocytes are referred as endogenous sources in the figure 1 itself they are pointed out as exogenous sources. Please clarify. In addition, in the figure legend the activation of cAMP is mentioned, but not shown in the figure itself, please clarify this more in the figure.

Response: We would like to thank the reviewer for their kind words. As suggested we have modified our title of Figure 1 to indicate that it refers to the endogenous and exogenous sources of ROS in sperm. We would also like to thank the reviewers for picking up our mistake regarding the source of ROS from leukocytes. We have now moved this to exogenous sources in text to match that of Figure 1. In addition we have added in cAMP into figure 1 as suggested.

  • Please check the consistent chemical formula for superoxide in the text and all figures. In figure 2 it is referred as “O2.-“ in the text it is “O2-“ Please make this consistent.

Response: We would like to thank the reviewers for pointing out this omission we have now modified in both the text and in the figures/figure legends for superoxide to O2-  .

  • Please check the reference in the text to figure 3, as Figure 3B is not referred to in the text. In addition, MitoSox Red is a specific mitochondrial superoxide indicator. Therefore please specify this circumstance in the text and figure legend.

Response: We have modified the text to include referencing to both Figure 3 as a whole and figure 3b specifically. We have also indicated both in text and in the figure legend that MSR is a specific to superoxide generated from mitochondria as suggested by the reviewer.   

Round 2

Reviewer 1 Report

Strongly suggest for publishing.